
# Particulate matter air pollution offsets ozone damage to global crop production

Luke D. Schiferl[1] and Colette L. Heald[1,2]

[1]Department of Civil and Environmental Engineering, Massachusetts Institute of Technology, Cambridge, Massachusetts, USA
[2]Department of Earth, Atmospheric and Planetary Sciences, Massachusetts Institute of Technology, Cambridge, Massachusetts, USA

*Correspondence to*: Luke D. Schiferl (schiferl@mit.edu)

**Abstract.** Ensuring global food security requires a comprehensive understanding of environmental pressures on food production, including the impacts of air quality. Surface ozone damages plants and decreases crop production; this effect has been extensively studied. In contrast, the presence of particulate matter (PM) in the atmosphere can be beneficial to crops given that enhanced light scattering leads to a more even and efficient distribution of photons which can outweigh total incoming radiation loss. This study quantifies the impacts of ozone and PM on the global production of maize, rice, and wheat in 2010 and 2050. We show that accounting for the growing season of these crops is an important factor in determining their air pollution exposure. We find that the effect of PM can offset much, if not all, of the reduction in yield associated with ozone damage. Assuming maximum sensitivity to PM, the current (2010) global net impact of air quality on crop production is positive (+6.0 %, +0.5 %, and +4.9 % for maize, wheat, and rice, respectively). Future emissions scenarios indicate that attempts to improve air quality can result in a net negative effect on crop production in areas dominated by the PM effect. However, we caution that the uncertainty in this assessment is large due to the uncertainty associated with crop response to changes in diffuse radiation; this highlights that more detailed physiological study of this response for common cultivars is crucial.

## 1 Introduction

Exposure to air pollution leads to over 4 million premature deaths per year (Cohen et al., 2017) and can impact the growth of vegetation (Avnery et al., 2011; Mercado et al., 2009). At the same time, pressure on food production continues to rise with increasing global population. The close proximity of human population to crop production areas means that anthropogenic influences on air quality (defined here as ozone and particulate matter (PM)) have had and will continue to have an impact on our ability to adequately feed a growing human population. As of 2014, the United States (US), Canada, Europe, India, and China make up 52 % of the global population and together are responsible for 72 %, 76 %, and 51 % of global maize, wheat, and rice production, respectively (FAOSTAT, 2017; United Nations, 2017). In China, which accounts for 19 % of the global population, rapid industrialization contributes to frequent air quality problems (Guo et al., 2014). These air quality concerns



occur alongside intense food production schemes, where China's crop production (21 %, 17 %, and 28 % of global maize, wheat, and rice production, respectively) often exceeds the county's population proportion. In less developed countries, such as India, where population is expected to increase by more than 25 % by 2050, food-related stress may accompany the push to industrialize. Given the importance of these intense food production areas, it is vital to quantify the relevant air quality impacts

on crop growth in order ensure proper guidance for future air quality and agricultural policy.

Surface-level ozone ($O_3$), is formed from the oxidation of carbon monoxide (CO) or volatile organic compounds (VOCs) in the presence of nitrogen oxides ($NO_x = NO + NO_2$) and sunlight. Ozone has a negative impact on crop production by reducing gas-exchange and inflicting phytotoxic damage on plant tissues (Lombardozzi et al., 2012; Sitch et al., 2007; Wilkinson et al.,

2011). The crop-specific relationships between surface ozone concentration and crop yield loss have been established using several exposure metrics. These metrics either account for mean exposure (e.g., M12) or cumulative exposure over a threshold concentration (e.g., AOT40). Observations relate the metrics for ozone exposure to relative yield (RY) for a specific crop (Adams et al., 1989; Lesser et al., 1990; Mills et al., 2007). For example, wheat is found to be more sensitive to ozone damage than maize and rice. These relationships have been used by previous studies to perform global crop-damage assessments. Van

Dingenen et al. (2009) find present day (2000) RY losses due to ozone to be 3–5 % for maize, 7–12 % for wheat, and 3–4 % for rice, with ranges due to variation in exposure metric. Shindell et al. (2011) show that future (2000 to 2030) changes in vehicle emissions will reduce RY losses due to ozone for all crops in North America and Western Europe, while the opposite is expected in India and China. Similar methodology is applied by Tai et al. (2014) to examine the confounding effects of ozone pollution control and climate change on crop production.

Much less studied is the effect of PM on crop production. PM can be directly emitted (e.g., mineral dust) or formed through chemical processes (e.g., sulfate). The make-up of PM in a particular location is dependent on its source and chemical environment. PM scatters or absorbs light, reducing the total shortwave (SW) light which reaches the surface, but light scattering also increases the diffuse fraction (DF) of this light (SW = direct + diffuse, $DF = \frac{\text{diffuse}}{\text{SW}}$). Whereas direct light only

reaches the top leaves of the plant (which can become light-saturated), an increase in diffuse light allows the radiation to penetrate to lower levels (Kanniah et al., 2012). The overall change (both sign and magnitude) in crop productivity from these competing effects (SW v. DF) depends on local light conditions and crop type. For example, crops with a $C_3$ photosynthetic pathway (e.g., wheat) are much more likely to become light-saturated than $C_4$ crops (e.g., maize) (Chapin et al., 2002). Depending on the saturation levels of both the sunlit and shaded leaves of a plant, a reduction in SW from full sunlight would

affect $C_4$ plants more, but an increase in DF to shaded leaves would produce a greater increase in $C_3$ plant productivity than for $C_4$ plants. Furthermore, enhanced light scattering under cloudy skies diminishes the PM impact, leading to seasonal and regional differences in the relative impact of PM on crop growth.



Greenwald et al. (2006) modify an existing process-based crop model to incorporate effects from scattered and diffuse light. This study uses offline meteorology and specified aerosol optical depth (AOD) at specific sites to quantify the effect of PM on yields of maize, wheat, and rice. They find relationships between yield and AOD for each crop at each location by relating the DF to the radiation use efficiency (RUE). RUE describes how well the plant uses available light, accounting for physiological

and environmental differences. They modify the RUE in their simulation as a function of DF according to various possible levels (e.g., max ΔRUE = 0 %, 50 % or 100 %), following from the theory of Sinclair et al. (1992). Under the most realistic scenario at various sites, the effect of PM on yield was found to be –10 to 0 % for maize, –5 to +5 % for wheat, and –10% to +10 % for rice. This assumes that the RUE of maize ($C_4$) is less sensitive to DF (max ΔRUE = 0–50 %) than the RUE of wheat and rice ($C_3$) (max ΔRUE = 50–100 %). The relationship between DF and RUE is highly uncertain, both in magnitude (max

ΔRUE) and in shape. For example, Rochette et al. (1996) observe a linear relationship over a maize field, rather than the hyperbolic relationship used in Greenwald et al. (2006). These various DF-to-ΔRUE relationships are shown in Fig. 1. The enhancing effect of aerosol diffuse light on plant productivity, related by AOD, has also been observed by Cirino et al. (2014) and Strada et al. (2015). However, these relationships are not easily translatable to DF and RUE values, and it is difficult to remove the impacts of clouds from such observations.

Previous global modeling studies have focused on the effects of diffuse light on total carbon or net primary productivity. For example, Mercado et al. (2009) quantify the spatially-varying effects of aerosols on carbon flux and productivity as a whole using a land-surface scheme along with AOD from a separate chemistry simulation, but they do not focus on crop production. They show that the growing global diffuse fraction increased the land carbon sink by 25 % between 1960 and 1999. Matsui et

al. (2008) study the effect of aerosol light scattering on photosynthesis on a regional level with high spatial resolution using a land surface model. They find the effect to be largest (and positive) at noontime under cloudless conditions, but less over croplands than over forests due to a lower leaf area index (LAI) in these regions. Previous studies have also looked at how ozone and/or PM impact natural vegetation and the carbon cycle, often by incorporating more advanced canopy or leaf-scale process modeling (Strada and Unger, 2016; Yue et al., 2017; Yue and Unger, 2014). Our study is the first to contrast the large-

scale impact of ozone and PM on global managed vegetation (crops).

This study uses the GEOS-Chem chemical transport model to simulate ozone and PM along with the Rapid Radiative Transfer Model for GCMs (RRTMG) to simulate PM's impact on radiation. We then use existing relationships from the literature to quantify the effects of both ozone and PM on crop production globally under both current (2010) and future (2050) emissions

scenarios. We contrast only the light absorbing and scattering effects of PM with the negative impacts of ozone and do not consider secondary feedbacks associated with clouds or reduced radiation reaching the surface (e.g., hydrology, temperature).



## 2 Tools

### 2.1 GAEZ Crop Production

The base crop production used in this study comes from the Global Agro-Ecological Zones (GAEZ) assessment for 2000 (www.fao.org/nr/gaez), developed by the Food and Agriculture Organization (FAO) of the United Nations along with the

5 International Institute for Applied Systems Analysis (IIASA).We scale the 2000 base production to 2010 values determined by the country-level trend between 2000 and 2010 from the FAO (www.fao.org/faostat). GAEZ crop production information is available at $5' \times 5'$ horizontal resolution and we maintain this high spatial resolution when adjusting production by the relative air quality effects calculated on the GEOS-Chem grid.

We consider three staple crops: maize, wheat, and rice. In 2010, these made up over 90 % of global cereal caloric intake and accounted for over 40 % of global total caloric intake (not including animal products) (FAOSTAT, 2017). The global baseline crop production is 871 Tg for maize, 667 Tg for wheat, and 705 Tg for rice according to GAEZ values scaled to 2010. Maize production is largest in the US+Canada region, accounting for 37 % of global production. China+Southeast (SE) Asia follows with 23 % of global maize production. Wheat production is greatest in Europe and makes up 31 % of global production, while

China+SE Asia and India hold about a 15 % share of wheat production each. China+SE Asia and India dominate rice production, with 44 % and 33 % of the global total, respectively.

We estimate the impact of air pollution on crops globally, however we note the domain of our figures focuses mainly on the industrialized, developed regions of the northern hemisphere mentioned above. Over three-quarters of our crops (maize, wheat,

and rice) are grown in this domain. On average, southern hemispheric air quality is cleaner. Food crops in those regions are also more varied (e.g., pulses in Africa), and their response to environmental stress is not as well understood. We neglect soybean production, because a metric relating potential carbon production to SW and DF consistent with the other crops (see Sect. 3.2) is not available.

### 2.2 GEOS-Chem Simulation

#### 2.2.1 General Description

We simulate emissions, chemistry, and wet and dry deposition processes relevant to ozone and PM concentrations in the troposphere in three dimensions using v10-01 of the GEOS-Chem chemical transport model (www.geos-chem.org). The model is driven by GEOS-5 meteorology from the NASA Global Modeling and Assimilation Office (GMAO) and is run globally at $2° \times 2.5°$ horizontal resolution with 47 vertical hybrid sigma layers for 2009 and 2010. Model time steps are set to 15 min for

transport and convection and 30 min for emissions and chemistry. GEOS-Chem contains sulfate-nitrate-ammonium thermodynamics coupled to an ozone–VOC–NO$_x$–oxidant chemical mechanism (Park et al., 2004; Pye et al., 2009). ISORROPIA II partitions ammonium nitrate between the gas and particle phase (Fountoukis and Nenes, 2007). The wet



deposition scheme in the model is described by Liu et al. (2001) for aerosols and by Amos et al. (2012) for gases, and the dry deposition processes are described by Wang et al. (1998) and Zhang et al. (2001).

Global anthropogenic emissions of $NO_x$, carbon monoxide, and sulfur dioxide ($SO_2$) come from the Emission Database for
Global Atmospheric Research (EDGAR) v4.2 (edgar.jrc.ec.europa.eu). The global Reanalysis of the TROpospheric chemical composition (RETRO) inventory is used for anthropogenic NMVOC emissions (Hu et al., 2015a), with global anthropogenic and natural ammonia ($NH_3$) emissions from the Global Emission Inventory Activity (GEIA) inventory. Biofuel emissions follow Yevich and Logan (2003). For anthropogenic (and in some cases biofuel) emissions, regional inventories overlay these global inventories in the US (National Emissions Inventory for 2011 (NEI-2011) v1 implemented by Travis et al. (2016)),
Canada (Criteria Air Contaminants (CAC) inventory (www.ec.gc.ca/pdb/cac/cac_home_e.cfm)), Mexico (Big Bend Regional Aerosol and Visibility Observational (BRAVO) Study Emissions Inventory (Kuhns et al., 2005), Europe (European Monitoring and Evaluation Programme (EMEP) (www.ceip.at)), and East Asia (MIX Asian emissions inventory (Li et al., 2014)). We modify NEI-2011 by reducing non-electric generating unit (non-EGU) $NO_x$ emissions by 60 % as suggested by Travis et al. (2016). Black carbon (BC) and organic carbon (OC) from anthropogenic sources are emitted globally, described
by Bond et al. (2007) and implemented by Leibensperger et al. (2012). Global biomass burning emissions come from the Global Fire Emissions Database v4.1 (GFED4) (van der Werf et al., 2017). Dust emissions are described by Fairlie et al. (2007), and sea salt emissions are described by Jaeglé et al. (2011). Lightning $NO_x$ emissions are from Murray et al. (2012), soil $NO_x$ emissions are from Hudman et al. (2012), and biogenic VOC emissions are from the Model of Emissions of Gases and Aerosols from Nature (MEGAN) v2.1 from Guenther et al. (2012) and implemented by Hu et al. (2015b). To account for
secondary organic aerosol (SOA), ten percent of monoterpene emissions by carbon are added to OC emissions as done by Park et al. (2003).

We output the hourly surface ozone concentration for use in quantifying the ozone impact on crop production. The ozone concentration in the surface grid box, nominally 120 m deep, is scaled to a 1 m canopy height using the simulated aerodynamic
resistance and dry deposition velocity for cropland. This method for accounting for the near-surface concentration gradient is described by Zhang et al. (2012) and has been previously implemented for ozone by Lapina et al. (2016) and Travis et al. (2017). Hourly PM concentrations at all vertical levels are read into RRTMG for calculation of their radiative impacts. In our simulation, PM refers to the sum of all simulated aerosol species: sulfate ($SO_4^{2-}$), nitrate ($NO_3^-$), ammonium ($NH_4^+$), BC, OC, sea salt, and dust.

**2.2.2 RRTMG**

RRTMG (Iacono et al., 2008) uses the correlated-k method to quickly calculate the atmospheric radiation flux throughout the vertical column and was implemented online into GEOS-Chem by Heald et al. (2014), together referred to as GC-RT. RRTMG simulates extinction from water vapor, ozone, greenhouse gases, aerosols, clouds and Rayleigh scattering over 16 longwave





and 14 SW bands. In GC-RT, ozone and aerosol concentrations are simulated in GEOS-Chem, greenhouse gas concentrations are prescribed from climatology, and water vapor concentration and cloud properties are taken from the GEOS-5 assimilated meteorology. A log-normal size distributed bulk scheme is used for all aerosols (2 bins for sea salt, 4 for dust), and consistent optical properties have been set in GC-RT. Using GC-RT, we output hourly downward SW radiation as well as the diffuse

fraction at the surface, both with and without PM under simulated real-time cloudiness (all-sky) conditions. In our analysis, we consider the radiative impacts of PM as a whole, and do not separate the impacts of individual particle types. For example, considering absorbing BC alone would result in only a small reduction in crop production due to SW loss and no enhancement in DF.

### 2.2.3 Evaluation with Observations

We compare the mean daytime GEOS-Chem surface ozone concentrations (scaled to 4 m height) with observations from the Air Quality System (AQS) network (www.epa.gov/aqs) in the US during summer (JJA) 2010. We find that the model is biased high by ~8 ppb on average. This high bias is consistent with previous studies (e.g., Travis et al., 2016). When comparing observations from the EMEP network (www.emep.int) in Europe to GEOS-Chem during the same time period, the model is similarly biased high (~9 ppb on average). This comparison is shown in Fig. 2. The addition of halogen chemistry by Sherwen

et al. (2016) suggests that future versions of the model will have lower ozone concentrations. Similar publicly-available network surface ozone measurements are not available for India and China during this time period.

We also compare ozone and PM surface observations from a site at the Indian Institute of Science Education and Research (IISER) Mohali in Chandigarh, India provided by Sinha et al. (2015) and Pawar et al. (2015), respectively, with simulated

concentrations from GEOS-Chem (Fig. 3). While an exact comparison between our 2010 simulation and the 2011−2014 observations is not possible, our qualitative comparison suggests that the model is biased slightly high, but does a good job reproducing the seasonal cycle of ozone concentration, with elevated concentrations during the dry phase (October−June) and lower concentrations during the wet phase of the monsoon. However, the model is unable to reproduce the magnitude of decline in ozone concentration at the peak of the wet season (July−September). For PM, the model generally reproduces the

magnitude of the observed concentrations. The model fails, however, to capture the observed higher concentrations during the dry phase of the seasonal cycle, especially over the winter (November−February).

Finally, we compare the observed daytime hourly SW and DF at an AmeriFlux site over a maize field in Mead, Nebraska with those simulated parameters from GC-RT during JJA 2010 (doi:10.3334/ORNLDAAC/Daymet_V2). Overall, the magnitudes

of the observed parameters compare well with the model. By including PM and clouds in the calculation of the SW and DF in GC-RT, the model is better able to capture the observed radiation. For example, including clouds and PM results in the simulation of a range of lower SW values (slope reduces from 1.19 to 1.13) and higher DF values (slope increases from 0.45 to 0.91), more consistent with those observed.



## 3 Methodology

### 3.1 Ozone

Using the hourly surface ozone concentrations from GEOS-Chem, we calculate ozone exposure metrics over the final 92 days (roughly 3 months) of a growing season ending in 2010 as done by Tai et al. (2014). This growing season is determined by the

5 University of Wisconsin Center for Sustainability and the Global Environment (UW SAGE) global crop calendar containing the planting and harvest dates by crop species and variety (maize, spring wheat, winter wheat, and rice) (Sacks et al., 2010). The spring (28 % by mass, globally) and winter (72 %) wheat distribution at each location is taken from the crop planting dates used in the pSIMS/DSSAT crop model (Elliott et al., 2014). The global distribution of planting and harvesting dates used here is shown in Fig. 4. Although double-cropping does occur in some regions, such as the sub-tropics, our study assumes mutual

exclusivity at a given location. We calculate AOT40 (Eq. 1) for maize, wheat, and rice, M12 (Eq. 2) for maize, and M7 (Eq. 3) for wheat and rice:

$$AOT40 = \sum_{t=08:00}^{t=19:59} 10^{-3}([O_3]_t - 40) \tag{1}$$

$$M12 = \frac{1}{n} \sum_{t=08:00}^{t=19:59} [O_3]_t \tag{2}$$

$$M7 = \frac{1}{n} \sum_{t=09:00}^{t=15:59} [O_3]_t \tag{3}$$

where $[O_3]_t$ is the hourly surface ozone concentration in ppb, $t$ the time each day in the summation and listed in local time, and $n$ is the total number of hours in the growing season.

The global distribution of surface ozone using the M12 exposure metric is shown in Fig. 5 for each crop. This figure has been filtered for grid boxes with a baseline crop production (see Sect. 2.1) of greater than 0.01 Mg km$^{-2}$. M12 is generally higher over areas with large anthropogenic influence, including the US, Europe, India, and China. This is especially true for summertime crops, such as maize and rice, whose growing seasons correspond with higher ozone concentrations. Lower ozone concentrations occur during the winter wheat growing season period, and this seasonal contrast is particularly noticeable in

China, but also in the US. Each exposure metric is then related to RY using empirical relationships as listed in Table 1. Following Van Dingenen et al. (2009) and Shindell et al. (2011), we use the mean of these metrics for each species to calculate the total production change due to ozone.

### 3.2 Particulate Matter

We calculate the mean daily daytime (hours with SW > 0) SW and DF from the hourly GC-RT output, both with and without

PM under all-sky (including clouds) conditions. We sample these days to the entire crop calendar growing season ending in



2010 using the same calendar used for ozone. The mean change in SW and DF due to PM over the growing season for each crop is shown in Fig. 5. As expected, PM reduces SW radiation at the surface and increases the DF. The largest PM impacts (both for SW and DF) are over China. While similar in magnitude between seasons (crops) in northern China, the effect of PM is smaller on summer crops (maize and rice) in southern China due to increased cloud cover in that season. Under cloudy skies, PM has a proportionally smaller positive impact on DF compared to the negative impact on SW. However, both of these impacts are smaller under cloudy conditions compared to when the sky is clear. Similarly, cloudy conditions present during the wet monsoon season, the growing season for maize and rice in India, mask much of the impact of PM on radiation in this region. Conversely, during the dry season (winter wheat) cloud-free skies enhance the effect of PM on SW and DF. The PM impacts on radiation are comparatively small in magnitude in other regions, although areas are still significant in terms of total productivity.

The potential carbon production for a crop is calculated on a daily basis with and without PM using the daily SW and DF values and then summed over the growing season for the total potential carbon. Potential carbon is calculated following the DSSAT model for maize and wheat (Eq. 4) and rice (Eq. 5):

$$P_{carb} \propto 0.5 \times SW \times RUE_{s,DF} \tag{4}$$

$$P_{carb} \propto (0.5 \times SW)^{0.65} \times RUE_{s,DF} \tag{5}$$

where $P_{carb}$ is the potential carbon production and $RUE_s$ is crop-specific radiation use efficiency (given in Table 2). In the equations above, $RUE_s$ is modified according to various DF-to-$\Delta$RUE relationships. We calculate the effect of PM on crop production across three levels of impact: max $\Delta$RUE = 0 % (changes in SW only, direct effect), max $\Delta$RUE = 50 % (maximum 1.5 × RUE at DF = 0.8), and max $\Delta$RUE = 100 % (maximum 2 × RUE at DF = 0.8) based on literature values (Greenwald et al., 2006). Finally, we calculate the relative carbon production with PM compared to without PM for each crop under each relationship.

### 3.3 Relative Crop Production

To calculate the crop production change due to ozone and due to PM at each $\Delta$RUE relationship, we multiply the relative yield with ozone and the relative carbon production with PM, respectively, by global base production values for each crop from the GAEZ database as described in Sect. 2.1.

### 4 Results

### 4.1 Present-Day Impact of Air Pollution on Crops

The crop production changes due to air quality under current emissions (2010) are shown in Fig. 6. Ozone reduces crop production everywhere. This negative effect ranges from wheat, which is most sensitive to ozone damage, at –11.9 % to maize





at –4.4 % to rice at –3.4 % global production change, consistent with the results of Van Dingenen et al. (2009).The high wheat sensitivity to ozone damage is counteracted somewhat due to the lower winter ozone concentrations affecting winter wheat. Figure 7 shows regional crop production changes, and the ozone impact on wheat is consistently high in all regions. There is also a greater ozone impact on maize and rice production in China+SE Asia compared to other regions, due to high ozone

concentrations in this region (Fig. 5).

PM significantly enhances crop production throughout the globe when the diffuse effect is calculated using maximum potential sensitivity (max ΔRUE = 100 %, Fig. 6). Global crop production increases due to PM by +11.5 % for maize, +16.4 % for wheat, and +8.9 % for rice. Figure 7 shows that the PM effect on crop production is especially large in China+SE Asia and

India, regions with high PM concentrations. This is particularly dramatic for Indian wheat during the dry season with a gain of over 25 %.

When the ozone and PM effects are combined to estimate the total impact of air quality on crop production (Fig. 6), the negative impact of ozone is substantially mitigated by PM. In many regions, the net impact is positive, such as for maize in

China, northern US, and Europe, wheat in India, and rice in India and China. This indicates that the diffuse effect from PM outweighs that of ozone damage in these locations. The net global production change due to air quality in this case is +6.0 % for maize, +0.5 % for wheat and +4.9 % for rice. In this analysis, the ozone and PM effects are also calculated separately, and we do not account for compounding effects. Further, we do not examine the effects of PM on cloud formation or PM deposition onto plant surfaces.

While we show detailed results from one particular growing season, it is important examine how this magnitude may change from year-to-year. Thus, we use Modern-Era Retrospective Analysis for Research and Applications v2 (MERRA2) meteorology from the GMAO to simulate the impacts of air quality on crop production for 10 growing seasons (9 for wheat) for 2001–2010. We hold anthropogenic emissions constant from year-to-year, varying only those emissions affected by

meteorology. The 10 year range (min to max) of crop production change due to ozone alone, due to PM alone with max ΔRUE = 100 % and due to net air quality is plotted as bars in Fig. 7 around the standard base run for a growing season ending in 2010 previously discussed. This range in production change is small compared to the ozone effect alone and to the uncertainty in the ΔRUE relationship, which increases confidence that the results above are a robust representation of air quality impacts on crop growth beyond the 2010 growing season.

This analysis takes into account the total current ozone and PM from all sources. When isolating the impact of anthropogenic influenced air quality (since 1850) on crop production, we find that a significant portion of the PM effect (roughly half) is from natural sources (mainly dust). Nearly all of the ozone impact is anthropogenic in origin (influenced by the AOT40 threshold metric).





## 4.2 Uncertainty in the DF-to-ΔRUE Relationship

Given the high bias in our ozone simulation (see Sect 2.3.3 and Fig. 2), we contrast the ozone impact with the largest possible impact of diffuse light (max ΔRUE = 100 %) in Fig. 6. However, the relationship between DF and ΔRUE is highly uncertain. Figure 7 shows the span of possible impacts on crop production when considering the range of DF-to-ΔRUE relationships. For the case of max ΔRUE = 0 %, which can be referred to as the effect of direct radiation on PM on crop production, only the decrease in SW caused by PM is considered. As such, crop production decreases everywhere both when alone and when added to the impact of ozone damage. As the sensitivity to DF (max ΔRUE) increases, the impact of PM on crop production becomes more positive. For many regions and crops, this range is greater than the negative impact of ozone alone. The sensitivity of these results to the assumed response to diffuse radiation highlights the critical need for additional observational constraints on the response of crops to light. Further, the difference between $C_3$ and $C_4$ plants is not taken into account here since ΔRUE only varies by DF. Future work could make use of a canopy model to better predict the distribution of light onto sunny and shaded leaves and the resulting RUE of the plant.

The most realistic relationship value may be closer to the max ΔRUE = 50 % assumption, especially for maize, which is less sensitive to an enhancement from DF, in which case at least some significant negative ozone impact is offset by the diffuse PM effect. For completeness, we include in Fig. 7 the range of crop production change when a –10 ppb ozone concentration bias correction is applied both for ozone alone and for the net effect of air quality on crop production. While the absolute impact of air quality on crop production depends on the accurate simulation of ozone, PM, and the response of crop growth to these constituents, it is clear from these results that the PM impact on crop growth has the potential to be a major environmental factor in global food production.

## 5 Implications for Future Scenarios

Regional and global air quality is expected to evolve considerably in the coming decades in response to local air quality management policy (e.g., Riahi et al., 2011; Thomson et al., 2011). We estimate the impact of these changes on global crop production by repeating the single growing season analysis above using anthropogenic emissions (ammonia, sulfur dioxide, sulfate, nitric oxide, BC, OC, and VOCs) from based on two Representative Concentration Pathways (RCP) projections for 2050 (tntcat.iiasa.ac.at/RcpDb). The resulting change in crop production from 2010 to 2050 is shown in Fig. 8.

In the RCP 4.5 scenario, air quality is projected to improve (global average annual ozone and PM concentrations will decrease by about 10 % and 50 %, respectively) in all regions except India (Fig. 9). These improvements counteract each other, and there is little net impact on crop production for maize and rice, unless PM sensitivity to DF is less than estimated here (assuming max ΔRUE = 100 %). Production of wheat may increase by about 2 % given its higher sensitivity to the ozone clean-up



measures. In India, however, air quality degradation under RCP 4.5 may lead to net losses in wheat production, with about 2 % reduction expected.

Declines in crop production are more likely in the RCP 8.5 scenario. Under this scenario, ozone increases in most regions,
while anthropogenic PM continues to decline (Fig. 10). Globally, this leads to a small net total crop production loss for maize, wheat, and rice (<2 % decrease in production). As in RCP 4.5, both PM and ozone increase in India, with the net impact dominated by the deterioration in ozone air quality.

As in Fig. 7, ranges in estimated impacts corresponding to the uncertainty in the response of crop production to air quality are
10 shown in Fig. 8. In all cases, the impact of PM is reduced if a lower sensitivity to DF (max ΔRUE) is assumed.

Figure 8 demonstrates that future crop growth will be impacted by air quality management strategies. For example, a policy which reduces ozone concentrations at the surface would be beneficial by enhancing food production, especially in regions like China and India. In contrast, a policy which leads to a reduction in anthropogenic PM for the purpose of improving air
quality would have a negative impact on crop production. Given the impact of air pollution on global public health, mitigating human exposure should remain the top priority of air quality management. However, these results suggest that the impact of such policies on global food production should also be considered.

## 6 Conclusions

Previous studies have quantified the reduction in crop yields and associated economic costs based on surface ozone alone, but
it is imperative to understand all of the environmental impacts and limitations on crop growth given the pressure to enhance food production in the coming decades. This study broadens the study of environmental impacts on crop production by quantifying the impacts of both ozone and PM on current and future global crop production. We demonstrate that including the diffuse effect of PM on crop production can offset the negative impacts due to ozone. This offsetting nature of PM and ozone on crop production should feature in air quality management; future improvements in air quality may not be entirely
beneficial to crop production, as would be assumed when considering only the impact of ozone damage. Such a scenario may even cause a net negative impact on crop production. We further note that targeting reductions in specific aerosol types may have different effects on crop production (i.e., for absorbing v. scattering particles). The range of uncertainty regarding the relationship between diffuse radiation (DF) and the response of the crop (ΔRUE) is large and warrants further experimental study. More work is also needed to understand the timing of these effects during the growing season. Finally, it may be
important to consider how resource restrictions (e.g., limited water and nutrients) can impact these results.



**Acknowledgements**

Funding for this research was provided by the Martin Family Fellowship for Sustainability and the Abdul Latif Jameel World Water and Food Security Lab (J-WAFS) at MIT. The authors thank K. Travis for discussion of surface ozone corrections and the GEOS-Chem support staff and community for model documentation.

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





**Table 1. Relationships between ozone exposure metric and relative yield (RY) due to ozone.**

| Crop | AOT40 | M12/M7 |
|---|---|---|
| Maize | $RY = 1 – (0.00356 × AOT40)$ (Mills et al., 2007) | $RY = \exp[–(M12/124)^{2.83}]/\exp[–(20/124)^{2.83}]$ (Lesser et al., 1990) |
| Spring Wheat | $RY = 1 – (0.0163 × AOT40)$ (Mills et al., 2007) | $RY = \exp[–(M7/186)^{3.2}]/\exp[–(25/186)^{3.2}]$ (Adams et al., 1989) |
| Winter Wheat | Same as spring wheat | $RY = \exp[–(M7/137)^{2.34}]/\exp[–(25/137)^{2.34}]$ (Lesser et al., 1990) |
| Winter Wheat, China | $RY = 1 – (0.0228 × AOT40)$ (Wang et al., 2012) | |
| Rice | $RY = 1 – (0.00415 × AOT40)$ (Mills et al., 2007) | $RY = \exp[–(M7/202)^{2.47}]/\exp[–(25/202)^{2.47}]$ (Adams et al., 1989) |
| Rice, China | $RY = 1 – (0.00949 × AOT40)$ (Wang et al., 2012) | |

**Table 2. Base radiation use efficiency ($RUE_s$) for each crop.**

| Crop | $RUE_s$ [g C (MJ PAR)$^{-1}$] |
|---|---|
| Maize | 4.20 |
| Wheat | 2.70 |
| Rice | 5.85 |

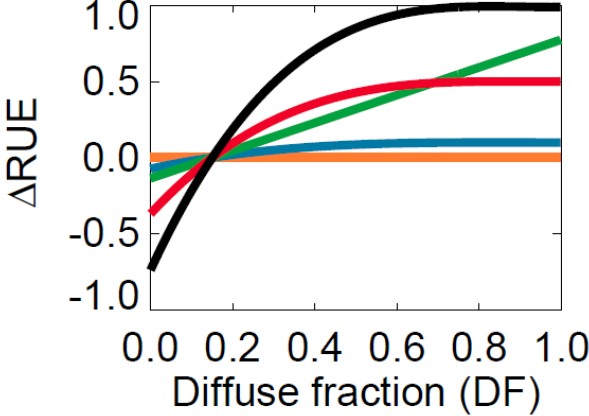

**Figure 1. Relationship between DF and ΔRUE for various assumptions: max ΔRUE = 0 % (orange), 10 % (blue), 50 % (red), and 100 % (black) from (Greenwald et al., 2006), and linear (green) from (Rochette et al., 1996).**





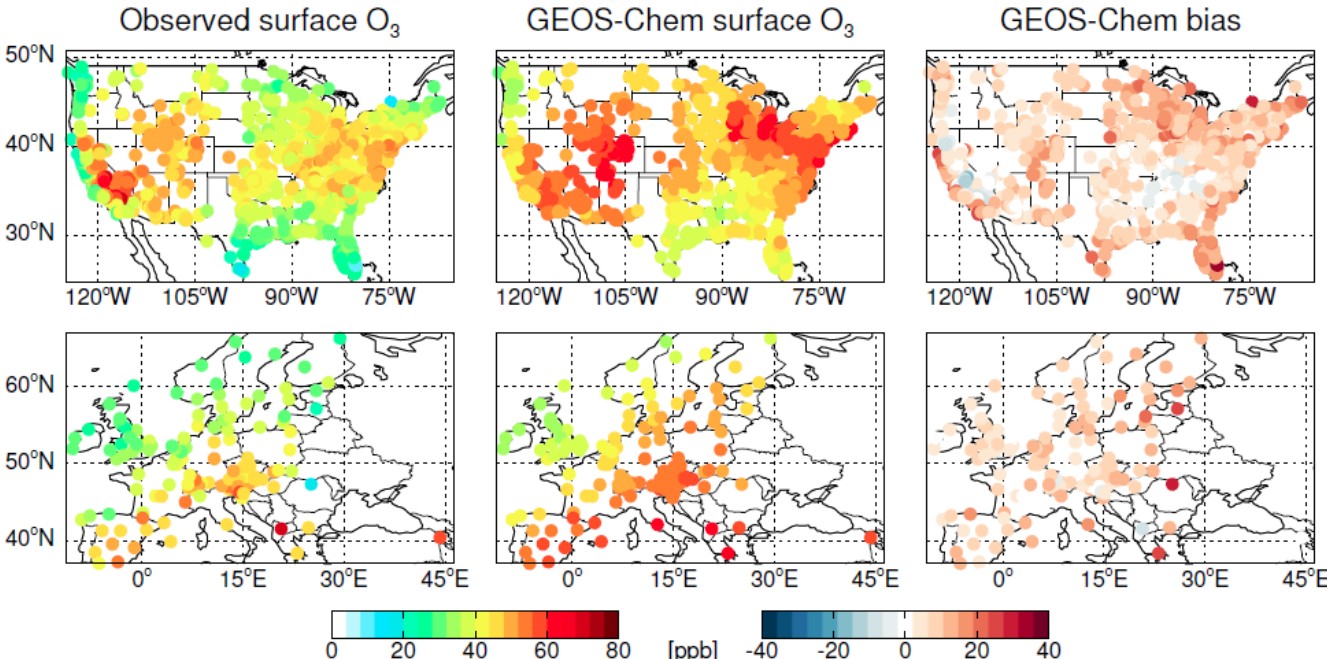

**Figure 2.** For both **(top row)** AQS network sites over the US and **(bottom row)** EMEP network sites over Europe: comparison of mean daytime (8:00–20:00 local time) **(left column)** observed and **(middle column)** GEOS-Chem simulated surface ozone concentrations for summer (JJA) 2010. Bias (simulated-observed) shown in right column.

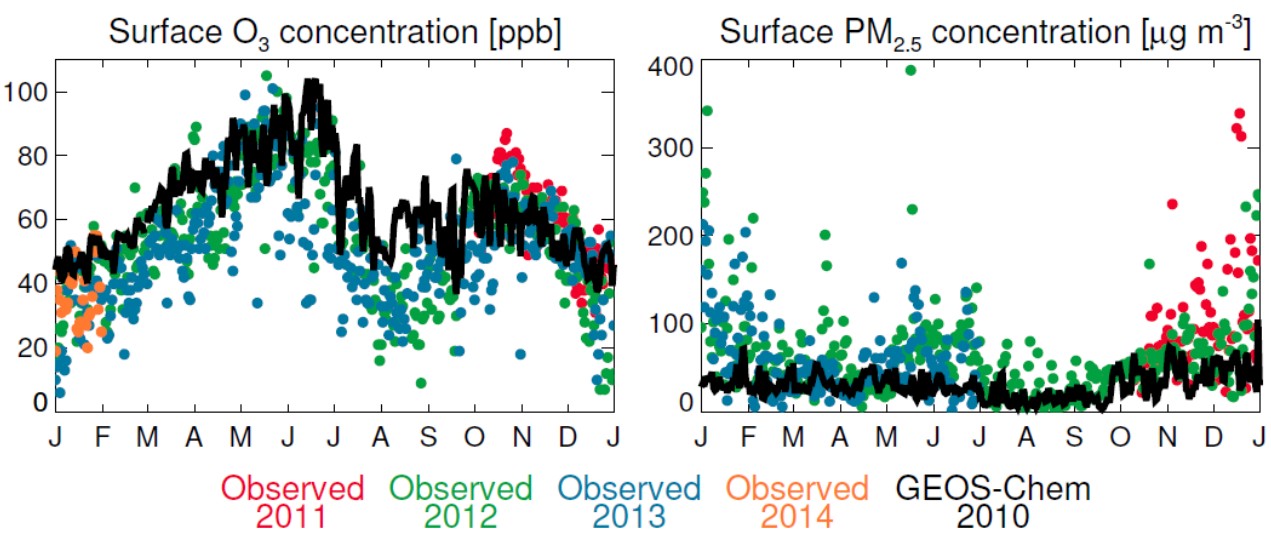

**Figure 3.** Daily surface **(left)** mean daytime (9:00–16:00 local time) ozone concentration and **(right)** mean daytime (12:00–16:00 local time) PM$_{2.5}$ concentration at Chandigarh, India. Observations in colored dots: 2011 (red), 2012 (green), 2013 (blue), and 2014 (orange). GEOS-Chem simulated values in black lines.




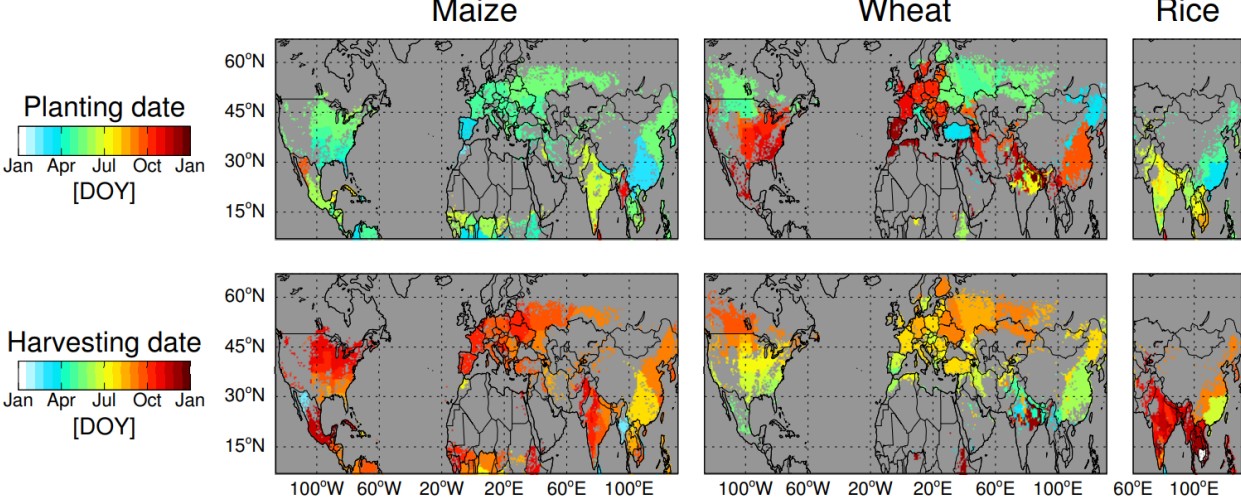

**Figure 4.** (top row) Planting dates and (bottom row) harvesting dates used in this study for (left column) maize, (middle column) wheat, and (right column) rice. Filtered for base crop production greater than 0.01 Mg km$^{-2}$.

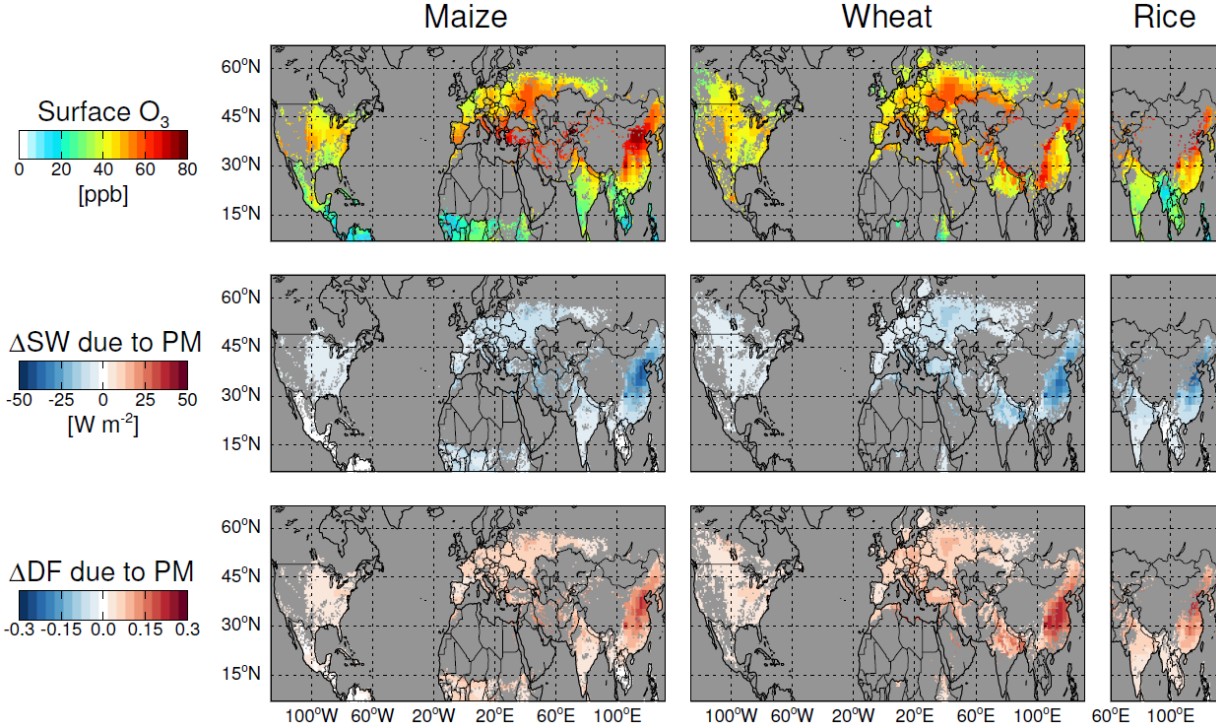

5   **Figure 5.** (top row) Mean daytime (8:00–20:00 local time) GEOS-Chem simulated surface ozone concentrations. Mean change in daytime (SW > 0) (middle row) downward SW radiation and (bottom row) DF of the SW radiation at the surface due to PM from GC-RT. Sampled to growing season ending in 2010 for (left column) maize, (middle column) wheat, and (right column) rice. Filtered for base crop production greater than 0.01 Mg km$^{-2}$.



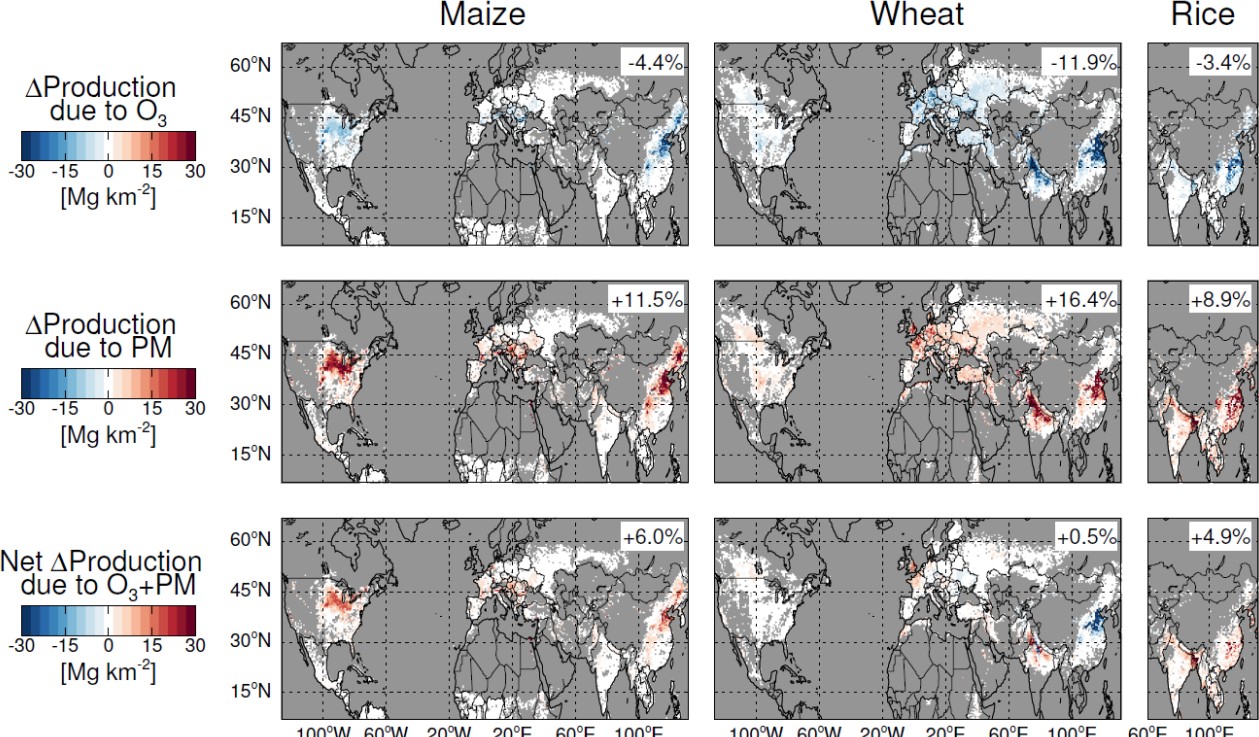

**Figure 6. Change in crop production due to (top row) surface ozone, (middle row) PM with max ΔRUE = 100 %, and (bottom row) both ozone and PM. Sampled to growing season ending in 2010 for (left column) maize, (middle column) wheat, and (right column) rice. Filtered for base crop production greater than 0.01 Mg km⁻². Global relative production change shown in upper right.**



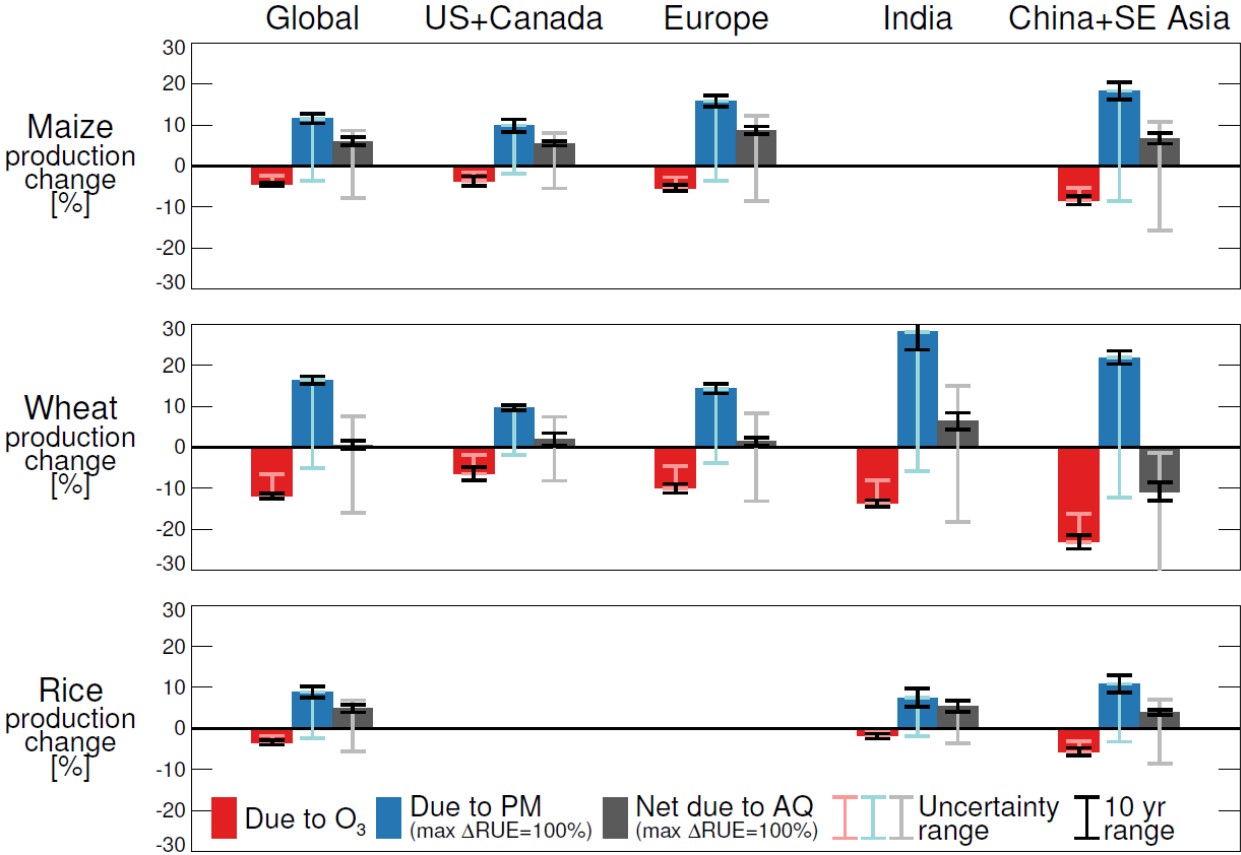

**Figure 7. Regional relative change in crop production due to surface ozone (red bars), PM with max ΔRUE = 100 % (blue bars), and both ozone and PM (gray bars). Sampled to growing season ending in 2010 for (top row) maize, (middle row) wheat, and (bottom row) rice. Light red, light blue, and light gray lines indicate range of production from 0 to –10 ppb surface ozone concentration correction, from max ΔRUE = 0 % to max ΔRUE = 100 %, and from both effects, respectively. Black lines indicate range of production change over 10 years of variable meteorology. Regions with a base production lower than 5 % of the global total are not shown.**




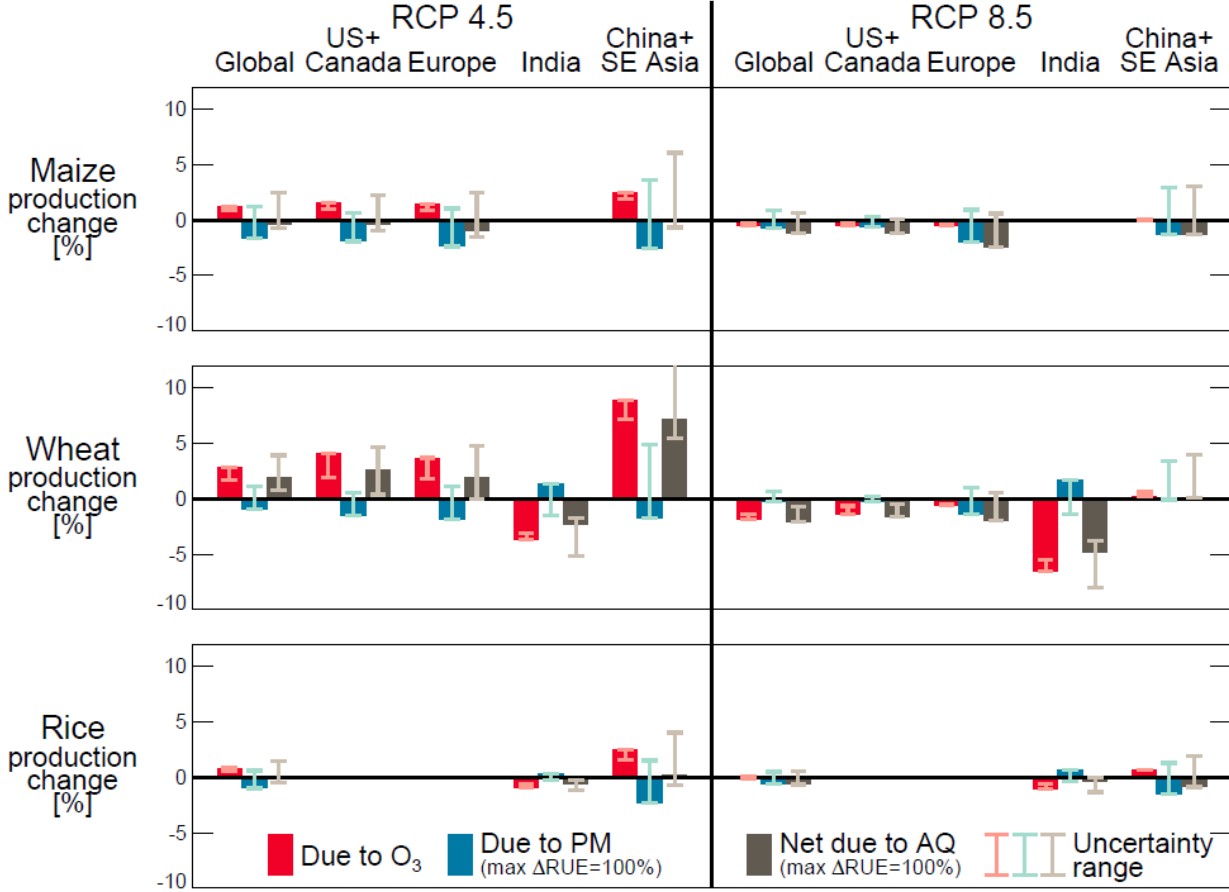

**Figure 8. For both (left) RCP 4.5 and (right) RCP 8.5 emissions scenarios: regional relative change in crop production due to surface ozone (red bars), PM with max ΔRUE = 100 % (blue bars), and both ozone and PM (gray bars). Change from 2010 to 2050 for (top row) maize, (middle row) wheat, and (bottom row) rice. Light red, light blue, and light gray lines are as in Fig. 7. Regions with a base production lower than 5 % of the global total are not shown.**




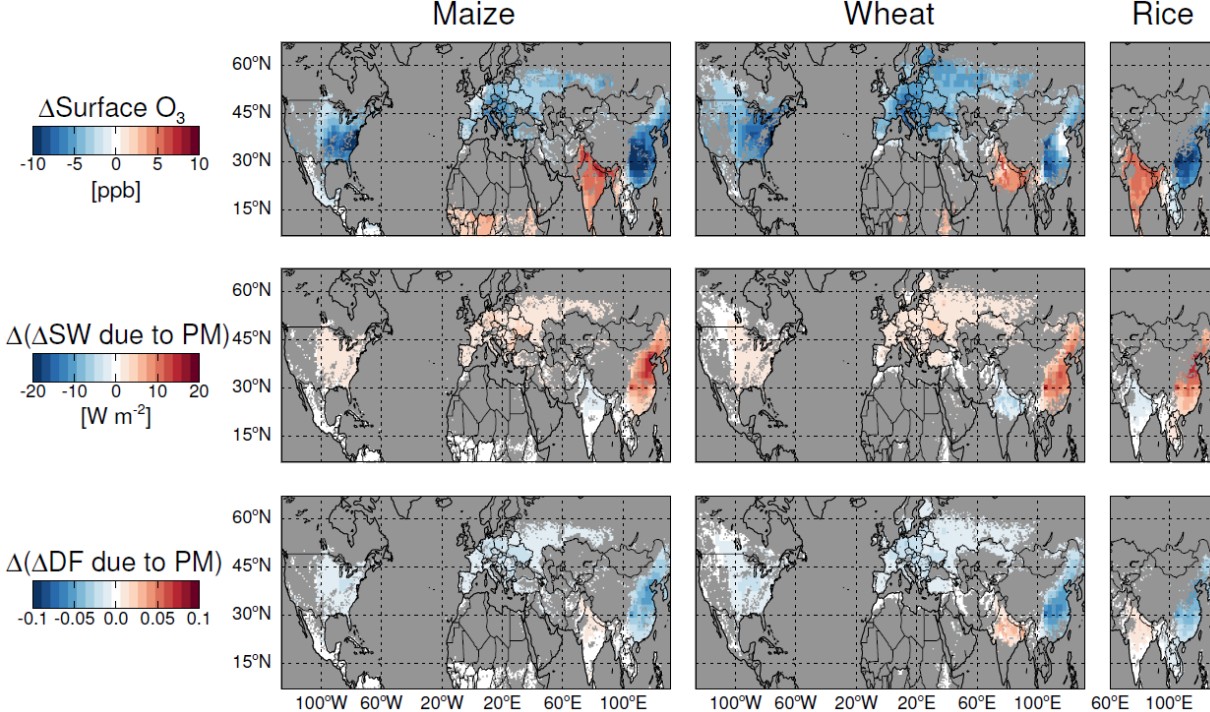

**Figure 9.** Same as Fig. 5, but for change in quantities using RCP 4.5 emissions scenarios (2050–2010).

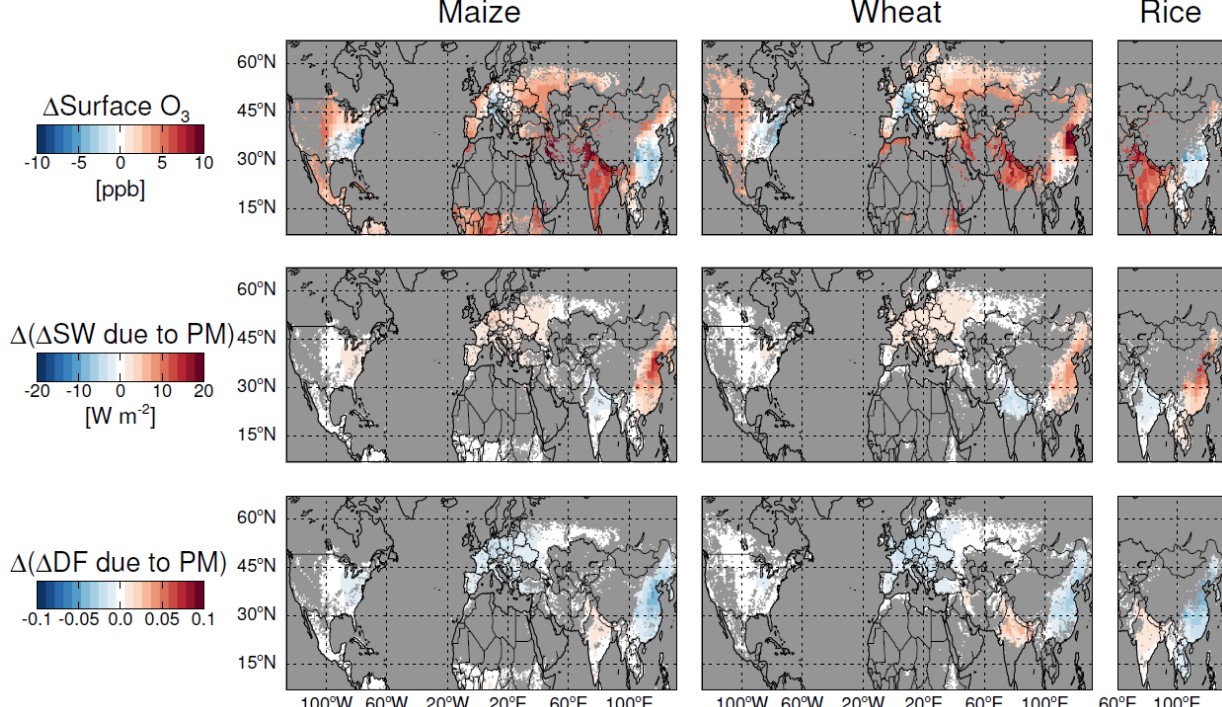

**Figure 10.** Same as Fig. 5, but for change in quantities using RCP 8.5 emissions scenarios (2050–2010).