# Peer review of "Particulate matter air pollution may offset ozone damage to global crop production"

_Atmospheric Chemistry and Physics, 2017_

## Short Comment (SC1) · 20 Jan 2018

A quick comment re the first paragraph of the Introduction section - Chameides et al. (Science, 1994) noted the proximity of crop production regions and anthropogenic emission regions in the NH mid-latitudes, and coined the term Continental-Scale Metro-Agro-Plexes (CSMAPs) to describe them.

---

## Referee Comment (RC1) · Anonymous Referee #1 · 23 Feb 2018

**Review of: *Particulate matter air pollution offsets ozone damage to global crop production**

By Luke D. Schiferl and Colette L. Heald

**Summary:**

The paper presents the results of a numerical modelling exercise which combines a base crop production model (GAEZ) with a chemistry transport model (GEOS-Chem). This framework is used to examine the response of managed vegetation (i.e. Maize, Wheat and Rice crops) exposed to different air quality assumptions (PD, RCP4.5 and RCP8.5). The modelling is done at the global scale but the reported results also discuss the main regions of production individually. Air quality here is characterized by Ozone and particulate matter (PM), the latter being represented by aerosol particles. Ozone is a pollutant with a well-known armful effect on plants physiology (i.e. it decreases their productivity). PM reduces the amount of global radiation available for plants photosynthesis (i.e. ir decreases productivity as well) but it also promotes higher level of diffuse radiation (i.e. it increases productivity). Overall the increase in diffuse fraction (DF) is assumed to enhance vegetation light use efficiency despite a reduction in global radiation. Considering both ozone and PM impacts on crop production, the authors claimed that PM offsets the ozone damaging effect for most of the regions and type of crops considered for present days emission levels (ca 2010). The authors also conduct additional experiments to evaluate the impact of future air quality policies assuming both RCP4.5 and RCP8.5 emission trajectories. They highlight that a reduction in pollution from particulate matter may result in a net negative effect on food production.

**General comments:**

Linking climate-composition with the terrestrial carbon is an active field of research. Few studies so far have considered the combined impact of PM and Ozone on the global carbon budget. Even lesser have analysed these impact on crop production specifically. The paper brings novel insight and help to appreciate the potential role of climate-composition on food production, highlighting interesting implication for air quality policies. This makes the paper absolutely relevant for publication in ACP.

The paper is well structured and reads nicely. The figures are clear and produced with great attention to details. I strongly support its publication in ACP after addressing these few comments.

**Specific comments:**

- Modelling the impact of aerosol on the radiative/energy budget and consequently on vegetation productivity is difficult and remains poorly constrained. Intermediate complexity models such as those used in the present study are a good step toward assessing these relationships. However, such framework may overlook some critical feedbacks (e.g. temperature change induced by the aerosol cooling which may shift/bring closer vegetation from its optimum productivity regime, change in cloudiness, change in the hydrology cycle / soil moisture, …). I would recommend softening the title which makes a strong statement, especially when considering that the main results reported in the abstract are based on the most sensitive DF assumption ($\Delta$RUE=100%) and that the framework is not fully coupled.

- As the authors acknowledge, the present study does not consider the secondary feedbacks associated with clouds or reduced radiation reaching the surface (e.g. hydrology, temperature). In a recent study, Unger et al. (2017) argue that the aerosol cooling impact over the Amazon drives a net primary productivity (NPP) increases that is 5–10 times larger than estimates of diffuse radiation fertilization by biomass burning aerosol in this region. These estimates may well be specific to the location, the type of biome considered and the numerical representation of the processes involved but would you believe that PM may then still be able to offset ozone damage in future climate if you were considering these cooling effects?

- Suggestion: To make an even stronger point in the abstract/conclusion, could you provide an estimate of the associated economic cost ($$) attributed to the effect of AQ on crop production?

- In your RCP4.5 and RCP8.5 experiment, do you also modify the meteorology to be representative of 2050 or are you just modifying the PM emissions?

- Along the same lines, in 2050, do you assume the same Land Use (i.e. vegetation distribution) as in PD or do you consider a different distribution for crops?

- Has the growing season changed for the 2050 period?

- Do natural aerosols change in your future climate projections (e.g. Allen et al. 2016)?  Could that have an implication for assessing the impact of anthropogenic aerosols on crops?

- How does the representation of diffuse light fertilization used here compares with more mechanistic approach such as those used in CLM (Bonan et al., 2011), YIBs (Yue et al., 2017) or JULES (Mercado et al., 2009)? Is it particularly sensitive? Are crops more susceptible to DF than more vertically developed canopies?

- Same question about the representation of the Ozone damage. How does it compare with representation such as the one of Sitch et al. (2007)?

- Is the representation of the impact of DF on RY based on observations? If so, is it considered to be robust globally or was it derived for a specific type of plant/crops at specific location?

- How the ΔRUE function is applied? Is it function of DF that provides a multiplying factor to adjust RY? Does the reduction in radiation due to PM impact RY as well?

- Page 9 – L18-20 can move/add in methods. So, do I understand correctly that the impact of Ozone on Pcarb and the impact of PM on RY are calculated individually?

- Although there are properly defined in the method, state explicitly what the AOT40 and M12 acronyms stand for in the introduction.

- Page 6, L30: Comparison of SW/DF → add: not shown.

**References**:

Allen R. J., Landuyt W. and Rumbold, S. T., An increase in aerosol burden and radiative effects in a warmer world Nat. Clim. Change 6 269–74 (2016).

Bonan, G. B., Lawrence, P. J., Oleson, K. W., Levis, S., Jung, M., Reichstein, M., Lawrence, D. M., and Swenson, S. C.: Improving canopy processes in the Community Land Model version 4 (CLM4) using global flux fields empirically inferred from FLUXNET data, J. Geophys. Res., 116, doi:10.1029/2010JG001593, (2011).

Mercado, L. M., Bellouin, N., Sitch, S., Boucher, O., Huntingford, C., Wild, M., and Cox, P. M.: Impact of changes in diffuse radiation on the global land carbon sink, Nature, 458, 1014–1017, doi:10.1038/nature07949, (2009).

Sitch, S., Cox, P. M., Collins, W. J., and Huntingford, C.: Indirect radiative forcing of climate change through ozone effects on the land-carbon sink, Nature, 448, 791–794, (2007).

Unger, N., Yue, X. and Harper, K. L., Aerosol climate change effects on land ecosystem services, Faraday Discussion (2017).

Yue, X., Unger, N., Harper, K., Xia, X., Liao, H., Zhu, T., Xiao, J., Feng, Z., and Li, J.: Ozone and haze pollution weakens net primary productivity in China, Atmos. Chem. Phys., 17, 6073-6089, (2017).

---

## Referee Comment (RC2) · Anonymous Referee #2 · 26 Feb 2018

Over the last ten years or so it has become more and more evident that air pollution has a strong impact on the Earth's ecosystems with consequences for global ecosystem productivity and wellbeing. It also affects the efficiency with which the terrestrial vegetation acts as a carbon sink with obvious consequences for the climate. The impacts can be detrimental, as is the case with ozone, or they can be beneficial when for instance aerosols increase the diffuse fraction of PAR thereby increasing plant productivity. The interactions are complex and strongly depend on the prevailing environmental conditions such as for instance cloudiness, temperature or drought conditions. With an increasing global population the vulnerability of food crops is of special importance. This manuscript examines the impacts of pollution in the form of surface ozone and

[Figure]

PM_2.5 on the productivity of major food crops (maize, wheat, rice) under PD (2010) and FU (2050) conditions at the global domain. The study applies global models that are coupled in an offline manner (GEOS-Chem to simulate atmospheric chemistry and RRTMG to compute the atmospheric radiation flux). Conventional pollution exposure metrics such as AOT40, M12 and M7 for ozone and radiation use efficiency (RUE) for PM_2.5 are used to quantify the impacts, in the latter case empirical DF-to-ΔRUE relationships are used to calculate the change in potential carbon production (P_carb).

I think that this study's research question is very timely and important. In principal, the applied methods and models are appropriate but I also believe that the conclusions drawn from the modelling stretch the capabilities of the tools to their limits or maybe even beyond, but I will com back to this specific point later in my review. Overall, I think the study is executed well, the manuscript is well written and logically consistent, the data and figures are adequate and support the principal findings. Thus I am satisfied that the paper can be published in AP albeit after the conclusions have been revised.

My main concern with this manuscript is with its conclusions. The authors state for instance that they "demonstrate that including the DFE of PM on crop production can offset the negative impacts due to ozone". I would prefer that it is made clear that this conclusion is drawn from a modelling study, that the bulk of the assessment is done with maximum DFE strength assumed and that feedbacks of PM with the climate system has been neglected (e.g., aerosol indirect effects, surface cooling, water vapour exchange, etc.). True, these facts have been mentioned individually in the preceding text but they have been omitted in the conclusions. I wish the conclusions were presented with more caution regarding the uncertainties.

In my reading of the paper I think the author have made a very valuable contribution by showing how large the uncertainties still are and that there is a potential for the DFE to counter the ozone impact but with current understanding the DFE can be almost anywhere between 0% and 100% of the ozone impact. For instance, Yue et al. in their study of the DFE over China (2017, doi:10.5194/acp-17-6073-2017) that the

ozone impact on productivity seems to dominate in their modelling study. The direct DFE accounted only for roughly 50% of the ozone impact and was further reduced to approximately 25% of this effect when taking into account the feedbacks of aerosols with clouds etc. I guess I am arguing that our understanding of the processes involved is still too poor to make strong statements that are not caveated.

My recommendations are to revise the conclusions and include caveats that I have pointed out above. I think even with those stronger caveats the paper is a very important and valuable contribution to the field. It is very well written and presented and needs no further revisions than the ones I pointed out.

I therefore recommend publication with minor revisions and leave it to the editor to "enforce" them, i.e., to decide if and to what extent the conclusions need to be revised.

---

## Author Comment (AC1) · 5 Apr 2018

Note: page and line references mentioned in author changes refer to positions within the revised manuscript below.

A quick comment re the first paragraph of the Introduction section - Chameides et al. (Science, 1994) noted the proximity of crop production regions and anthropogenic emission regions in the NH mid-latitudes, and coined the term Continental-Scale Metro-Agro-Plexes (CSMAPs) to describe them.

**Thank you for pointing this out Prasad! We have added this term on page 1, lines 25-26 and again on page 4, line 23.**

[revised manuscript text omitted]

---

## Author Comment (AC2) · 5 Apr 2018

Response to Anonymous Referee #1

Note: page and line references mentioned in author changes refer to positions within the revised manuscript below.

• Modelling the impact of aerosol on the radiative/energy budget and consequently on vegetation productivity is difficult and remains poorly constrained. Intermediate complexity models such as those used in the present study are a good step toward assessing these relationships. However, such framework may overlook some critical feedbacks (e.g. temperature change induced by the aerosol cooling which may shift/bring closer vegetation from its optimum productivity regime, change in cloudiness, change in the hydrology cycle / soil moisture, …). I would recommend softening the title which makes a strong statement, especially when considering that the main results reported in the abstract are based on the most sensitive DF assumption (ΔRUE=100%) and that the framework is not fully coupled.

**We agree with the reviewer that the reader may mistakenly interpret our title to suggest that all of the ozone damage is offset by PM. Given the uncertainties on the PM impact, we agree that a title change is appropriate. We have changed the title from "Particulate matter air pollution offsets ozone damage to global crop production" to "Particulate matter air pollution may offset ozone damage to global crop production"**

• As the authors acknowledge, the present study does not consider the secondary feedbacks associated with clouds or reduced radiation reaching the surface (e.g. hydrology, temperature). In a recent study, Unger et al. (2017) argue that the aerosol cooling impact over the Amazon drives a net primary productivity (NPP) increases that is 5–10 times larger than estimates of diffuse radiation fertilization by biomass burning aerosol in this region. These estimates may well be specific to the location, the type of biome considered and the numerical representation of the processes involved but would you believe that PM may then still be able to offset ozone damage in future climate if you were considering these cooling effects?

**The reviewer raises an interesting point. However, as indicated on page 4, lines 2-3, our simulations do not consider these effects (by design), and given the complex (and potentially regionally-varying) feedbacks, we are not able to speculate on the impacts. Given the reviewer's comment, we now re-iterate in the conclusions that we do not include these effects and that our exploration of sensitivity to air quality should not be viewed as a prediction:**

**Page 12, lines 5-8 *"Our study does not address the potential feedbacks of a changing climate on air quality; in addition to the direct air quality impacts discussed here, predictions of future crop production in any given region should consider these feedbacks along with local changes in environment and agricultural practices."***

• Suggestion: To make an even stronger point in the abstract/conclusion, could you provide an estimate of the associated economic cost ($$) attributed to the effect of AQ on crop production?

**We thank the reviewer for this suggestion. We did consider including such a calculation in this study (it was included in the first author's PhD thesis), but we determined that coupling the large uncertainties of the PM effect along with uncertainties in crop valuation techniques (variation in crop prices in time and space) led to highly uncertain economic cost estimates which were not particularly meaningful. We therefore prefer to not include these estimates here.**

• In your RCP4.5 and RCP8.5 experiment, do you also modify the meteorology to be representative of 2050 or are you just modifying the PM emissions?

**No, we do not modify the meteorology in the 2050 simulations. We only modify the anthropogenic emissions (both gases and PM). This has been clarified on page 11, lines 8-9.**

• Along the same lines, in 2050, do you assume the same Land Use (i.e. vegetation distribution) as in PD or do you consider a different distribution for crops?

**We assume the land use in the future 2050 simulations is the same as in the 2010 present day simulation. This has been clarified on page 11, lines 8-9.**

• Has the growing season changed for the 2050 period?

**No, the growing season has not changed for the 2050. We use the same crop calendar presented in Fig. 4. This has been clarified on page 11, lines 8-9.**

• Do natural aerosols change in your future climate projections (e.g. Allen et al. 2016)? Could that have an implication for assessing the impact of anthropogenic aerosols on crops?

**No, we do not change the natural aerosols. These are forced by constant meteorology and land maps as mentioned above. This has been clarified on page 11, lines 8-9.**

**We identify on page 10, line 12 that roughly half of the present day PM impact on crop production is due to natural sources (e.g., dust, sea salt, biomass burning). It is possible that changes to these natural aerosols may confound the changes in anthropogenic aerosols, but we have not investigated this or the impact of individual aerosol types on crops. Our goal in Sect. 5 was to demonstrate that future air quality management policy may have an impact on crop production, not provide a comprehensive prediction of changing aerosol and ozone impacts.**

• How does the representation of diffuse light fertilization used here compares with more mechanistic approach such as those used in CLM (Bonan et al., 2011), YIBs (Yue et al., 2017) or JULES (Mercado et al., 2009)? Is it particularly sensitive? Are crops more susceptible to DF than more vertically developed canopies?

**Our representation of diffuse light fertilization is more simplistic than that used in mechanistic models, but it is derived from separate theoretical direct and diffuse light pathways by Sinclair et al. (1992). Overall, crops (the focus of our study) have comparatively small canopies and are less sensitive to DF compared to the larger canopies of trees (which have been the focus of previous studies of this effect). This, plus the large uncertainty in the DF-to-ΔRUE relationship for crops, does not warrant the more complex calculation of the radiation pathways inside the canopies as done by the other models mentioned above. A portion of this explanation has been added on page 3, lines 28-31.**

• Same question about the representation of the Ozone damage. How does it compare with representation such as the one of Sitch et al. (2007)?

**As with the PM impact on diffuse light, our representation is simpler than the mechanisms presented by Sitch et al. (2007) which explicitly describe the impact of ozone on stomatal conductance and leaf damage. Instead, we use simple empirical relationships applied in previous studies which directly relate ozone concentration to crop yield, rather than compounding the multi-step process which results in a change in biomass, not crop yield. Such a physiological description for crops (linking ozone mechanistically to crop yield) is not available. A portion of this explanation has been added on page 3, lines 28-31. We also highlight the need to better understand the mechanistic connections between air quality and crop yield part of future work on page 12, lines 16-17.**

• Is the representation of the impact of DF on RY based on observations? If so, is it considered to be robust globally or was it derived for a specific type of plant/crops at specific location?

**The impact of DF on RUE is based on theory from Sinclair et al. (1992) as mentioned on page 3, line 7. We translate the change in RUE (ΔRUE) due to PM to RY using the relative change in potential carbon calculated with and without PM. This equation for RY has been added as Eq. 6 on page 8, line 29. This relationship is used globally, but we do not have the observations to support whether or not it is robust across regions. There may be regional variations such as those in the relationships between ozone concentration and RY (Table 1). These uncertainties are acknowledged in our Conclusions, where we highlight the need for additional experimental work (page 12, lines 13-15).**

• How the ΔRUE function is applied? Is it function of DF that provides a multiplying factor to adjust RY? Does the reduction in radiation due to PM impact RY as well?

**The potential carbon production is calculated using SW and RUE as in Eqs. 4 and 5, which has been adjusted from the DF (based on the DF-to-ΔRUE relationship), both with and without PM. The RY due to PM is then calculated using Eq. 6. The reduction in radiation is accounted for in changes to the SW term. This has been clarified on page 8, lines 21-30.**

• Page 9 – L18-20 can move/add in methods. So, do I understand correctly that the impact of Ozone on Pcarb and the impact of PM on RY are calculated individually?

**We moved this sentence to page 8, line 30. Yes the impacts of ozone and PM on crops are calculated individually, this is mentioned on page 9, lines 27-28.**

• Although there are properly defined in the method, state explicitly what the AOT40 and M12 acronyms stand for in the introduction.

**These definitions have been added on page 2, lines 12-14.**

• Page 6, L30: Comparison of SW/DF à add: not shown.

**This is added on page 7, line 7.**

Response to Anonymous Referee #2

Note: page and line references mentioned in author changes refer to positions within the revised manuscript below.

My main concern with this manuscript is with its conclusions. The authors state for instance that they "demonstrate that including the DFE of PM on crop production can offset the negative impacts due to ozone". I would prefer that it is made clear that this conclusion is drawn from a modelling study, that the bulk of the assessment is done with maximum DFE strength assumed and that feedbacks of PM with the climate system has been neglected (e.g., aerosol indirect effects, surface cooling, water vapour exchange, etc.). True, these facts have been mentioned individually in the preceding text but they have been omitted in the conclusions. I wish the conclusions were presented with more caution regarding the uncertainties.

In my reading of the paper I think the author have made a very valuable contribution by showing how large the uncertainties still are and that there is a potential for the DFE to counter the ozone impact but with current understanding the DFE can be almost anywhere between 0% and 100% of the ozone impact. For instance, Yue et al. in their study of the DFE over China (2017, doi:10.5194/acp-17-6073-2017) that the ozone impact on productivity seems to dominate in their modelling study. The direct DFE accounted only for roughly 50% of the ozone impact and was further reduced to approximately 25% of this effect when taking into account the feedbacks of aerosols with clouds etc. I guess I am arguing that our understanding of the processes involved is still too poor to make strong statements that are not caveated.

My recommendations are to revise the conclusions and include caveats that I have pointed out above. I think even with those stronger caveats the paper is a very important and valuable contribution to the field. It is very well written and presented and needs no further revisions than the ones I pointed out.

**Thank you for your review. We agree that the uncertainties in our study are high, and we have added caveats to the conclusions (Sect. 6) as were suggested.**

[revised manuscript text omitted]